# *IL-6* Polymorphisms Are Not Related to Obesity Parameters in Physically Active Young Men

**DOI:** 10.3390/genes12101498

**Published:** 2021-09-25

**Authors:** Ewelina Maculewicz, Bożena Antkowiak, Oktawiusz Antkowiak, Andrzej Mastalerz, Agnieszka Białek, Anna Cywińska, Anna Borecka, Kinga Humińska-Lisowska, Aleksandra Garbacz, Katarzyna Lorenz, Ewa Szarska, Monika Michałowska-Sawczyn, Łukasz Dziuda, Paweł Cięszczyk

**Affiliations:** 1Faculty of Physical Education, Jozef Pilsudski University of Physical Education in Warsaw, 00-809 Warsaw, Poland; ewelina.jask@gmail.com (E.M.); andrzej.mastalerz@awf.edu.pl (A.M.); katarzyna.lorenz@awf.edu.pl (K.L.); 2Military Institute of Hygiene and Epidemiology, 01-163 Warsaw, Poland; bozena.antkowiak@wihe.pl (B.A.); oktawiusz.antkowiak@wihe.pl (O.A.); aborecka@wihe.waw.pl (A.B.); eszarska@gmail.com (E.S.); 3Institute of Genetics and Animal Biotechnology of the Polish Academy of Sciences, Jastrzębiec, 05-552 Magdalenka, Poland; a.bialek@igbzpan.pl; 4Faculty of Biological and Veterinary Sciences, Nicolaus Copernicus University in Torun, 87-100 Torun, Poland; anna_cywinska@sggw.edu.pl; 5Faculty of Physical Education, Gdansk University of Physical Education and Sport, 80-336 Gdansk, Poland; kinga.huminska-lisowska@awf.gda.pl (K.H.-L.); monikamichalowska@op.pl (M.M.-S.); 6Warsaw University of Life Sciences—SGGW, 02-787 Warsaw, Poland; aleksandra_garbacz@sggw.edu.pl; 7Military Institute of Aviation Medicine, 01-755 Warszawa, Poland; ldziuda@wiml.waw.pl

**Keywords:** genetic polymorphisms, *IL6*, body mass index, fat percentage

## Abstract

Interleukin 6 (IL-6) is a cytokine with both pro- and anti-inflammatory actions, but is also considered as a “metabolic hormone” involved in immune responses, affecting glucose, protein and lipid metabolism. It has been proposed to be related to obesity, but various results have been presented. Thus, in this study, the very homogenous population of young, male military professionals, living in the same conditions involving high physical activity, has been selected to avoid the influence of environmental factors. The subjects were divided into groups depending on the obesity parameters BMI (body mass index) and fat percentage (fat%), and the following *IL-6* SNPs (Single Nucleotide Polymorphisms) were analyzed: rs1800795, rs1800796 and rs13306435. No relation was found between obesity parameters and *IL-6* polymorphisms rs1800795, rs1800796 and rs13306435. It may be postulated that even if a genetic predisposition involves *IL-6* genes, this effect in individuals with obesity of a low grade is minor, or can be avoided or at least markedly reduced by changes in lifestyle.

## 1. Introduction

Obesity is a growing and common metabolic disorder with the number of people suffering from it having almost tripled since 1975, according to the World Health Organization (WHO). The disease has been recognized as multifactorial, with a critical role of genetic and environmental factors [1,2]. The primary cause of obesity and excess weight is an energy imbalance between calorie consumption and expenditure. The simple mechanism is an increased intake of energy-dense foods (abundant in fat and sugars) and a decrease in physical activity due to the sedentary nature of many forms of work, changing modes of transportation and increasing urbanization. However, it has been proven that more than 40% of variations in body mass index (BMI) are related to genetic factors [3]. One hypothesis claims that some populations may carry genes that determine increased fat storage, which may be beneficial for the periods of starvation, but in developed societies results rather in obesity and diabetes mellitus type 2 [4]. Many candidate genes and polymorphisms have been considered, including variants of the interleukin 6 (*IL-6*) genes, involved in inflammatory process and many other mechanisms related to obesity [3]. 

IL-6 belongs to the IL-6 cytokine family, which also includes IL-11, IL-27, IL-31, leukemia inhibitory factor, oncostatin M, ciliary neurotrophic factor, cardiotrophin-1 and cardiotrophin-like cytokine. These cytokines exert their influence via transmembrane cell surface receptor complexes, being a dimer of low-cost polymer chains, combining the 130 kDa signal transduction protein gp130 (ß-chain) with a cell-specific factor. β-chains exist in two forms: as a membrane-bound receptor (mIL-6R) or as a soluble receptor (sIL-6R). This results in two different signaling pathways: mIL-6R signaling, called the classic signaling pathway, and sIL-6R signaling, known as the trans-signaling pathway [5]. IL-6, together with other cytokines from the same family, acting via mIL-6R, induces signal transduction though the activation of JAK tyrosine kinases. The sIL-6R plausibly regulates cytokine activity through the inhibition of IL-6 binding to the surface receptor, and also can serve as a carrier protein [6]. IL-6 is secreted by various cells, such as T and B lymphocytes as well as monocytes, but also by non-lymphoid cells such as fibroblasts, chondrocytes, osteoblasts, skeletal and smooth muscle cells as well as islet cells. The effect of IL-6 action depends on the activated pathway: the classic signaling pathway mediates regenerative or anti-inflammatory activities of IL-6, whereas trans-signaling pathway mediates the pro-inflammatory responses of IL-6 [5,7]. IL-6 promotes lipolysis in adipose tissue and suppresses the synthesis of lipids, thus decreasing blood lipid concentrations. Its activity is strictly related to lipid metabolism pathways, e.g., after injecting IL-6 into the liver of mice, PPAR increased and sterol regulatory element-binding protein-1c (SREBP-1c) decreased [5]. IL-6 also influences glucose tolerance through the negative regulation of visfatin. It may also antagonize the secretion of adiponectin, elevate triacylglycerol levels by enhancing the influence of gluconeogenesis and glycogenolysis and inhibit the effect of glycogenesis [4]. IL-6 also affects insulin resistance and mitochondrial activity, and exerts a significant role in vascular diseases, the neuroendocrine system and neuropsychological behavior. 

The *IL-6* gene is located on chromosome 7 at 7p21-p14, between D7S135 and D7S370; it consists of five exons and is 5 kb long. Its promoter includes numerous regulatory sites, allowing the induction of gene expression. Numerous single-nucleotide polymorphisms (SNPs) were found in the encoding sequence and in the promoter of the *IL-6* gene. One of the most commonly found and analyzed polymorphisms is the C>G transition at position -174 of the promoter (rs1800795) [6]. *IL-6*-related genes are involved in cell differentiation, proliferation and apoptosis. 

The pleiotropic effect of IL-6 described above has also been considered as related to obesity and both positive and negative associations between obesity and *IL-6* polymorphisms have been demonstrated [3]. Increased levels of IL-6 have been reported in obesity and obesity-related multifactorial diseases, such as diabetes, insulin resistance, cardiovascular disease, osteoarthritis and cancers [3,8,9,10,11]. Elevated blood levels of IL-6 in obese patients are related to the activity of adipose tissue, which secretes many pro-inflammatory cytokines (IL-1β, IL-6, TNF-α, inter-cellular adhesion molecule 1 (ICAM1), monocyte chemotactic protein 1 (MCP-1) and many adipokines, such as leptin and adiponectin) [5]. Up to 30% of IL-6 in the blood is secreted by adipose tissue, and its level positively correlates with BMI [6,12]. 

Till now, the exact relation between *IL-6* polymorphisms and the risk of obesity has not been clearly defined, and conflicting results have been presented [3]. We hypothesized that the selection of the study population, to minimize the effect of environmental and other factors, may be critical for the results of existing genetic studies. Thus the aim of this study was to investigate the most commonly studied polymorphisms of *IL-6* (rs1800795, rs1800796 and rs13306435) in a group of healthy young males, homogenous in terms of environment and lifestyle (military professionals), differentiated by body mass index—BMI—and the percentage of fat. 

## 2. Materials and Methods

### 2.1. Participants

Caucasian, unrelated, male military professionals (volunteers) were qualified for this genetic case-control study conducted in the years 2018-2019. They were randomly recruited via advertising on the Military University campus. All of the 125 participants were ancestrally fitted (they were Polish, their families having lived in Eastern Europe for 3 generations), aged 19–26. They did not differ in sex (males only), age or height (Table 1). The homogeneity of the population was ascertained by selection based on the questionnaire screening for exclusion criteria, such as past diseases, injuries and related ailments as well as the presence of severe and chronic pain of any organ or system, both in the past and currently. Their good health was confirmed by general medical examinations that included electrocardiography (ECG). All participants had similar levels of physical effort exposure due to their professional daily schedule, related to their military service and responsibilities. They lived in a hall of residence on the university's premises and ate the same meals at the students’ cafeteria. All the participants received an information sheet concerning the research details, the purpose and procedures of the study in addition to the potential risks and benefits associated with participation in this study. All participants gave their written consent for anonymous genotyping, the results of which would be confidential.

### 2.2. Anthropometric Analyses

Anthropometric measurements and body composition were obtained by routine methods. A portable stadiometer was used for measuring height without shoes (precision of 0.1 cm, TANITA HR-001, Tanita Corporation, Japan). A bioelectrical impedance analysis (BIA) using the TANITA MC-780 machine (Tanita Corporation, Japan) was used to measure body composition (including fat%) and body weight with an accuracy of 0.1 kg according to the instruction manual (lightly dressed and without shoes). The criteria set out by the WHO [13] were used in the assessment of BMI values. The following formula was used to calculate the BMI values: body mass index (BMI) = body weight/height^2^ (kg/m^2^). All measurements were made in accordance with the procedure specified in the manual and without any metal objects such as jewelry or keys. 

The subjects (*n* = 125) were divided into groups depending on BMI and fat percentage (fat%). Each parameter gave the differentiation into the control group (CON) or the overweight group (OVER). Table 1 shows detailed characteristics of the groups.

People with BMI values below 25.0 were classified into the control group CON_BMI_ (*n* = 83). People with a BMI of ≥ 25.0 were classified into the overweight group (OVER_BMI_, *n* = 42). People with fat% greater than 20% were classified into the OVER_Fat%_ group (*n* = 19), while people with fat% lower than 20% were classified into the CON_Fat%_ group (*n* = 106) [13,14]. 

### 2.3. Genetic Analyses

Copan FLOQSwabs (Copan Diagnostics, Inc., Murrieta, CA, USA) were used to collect the buccal cells donated by the subjects (each subject received two swabs). A High Pure PCR Template Preparation Kit (Roche Diagnostics, Basel, Switzerland) was used in order to extract the genomic DNA from the buccal cells. The extraction was done in accordance with the instructions provided by the manufacturer. Good quality and quantity DNA samples were stored at a temperature of −20 °C for further analysis. The exclusion criteria for further analysis were incomplete basic information, DNA extraction failure, DNA degradation and abnormal gene detection results. All samples were genotyped in duplicate on a CFX Connect Real-Time PCR Detection System (BioRad, Hercules, California, United States). The following TaqMan pre-designed SNP Genotyping Assays were used: *IL-6* (rs1800795) C___1839697_20, *IL-6* (rs1800796) C__11326893_10 and *IL-6* (rs13306435) C__31310956_10 single-nucleotide polymorphisms (SNPs) (Applied Biosystems, Waltham, MA, USA), which included primers and fluorescently labeled (VIC and FAM) MGBTM probes for allele detection. TaqPath™ ProAmp™ Master Mix (Applied Biosystems, Waltham, MA, USA) was used for genotyping, according to the manufacturer’s protocol. Briefly, the conditions for reaction were as follows: 30 s of pre-read at 60 °C, 5 min of initial denaturation at 95 °C, 40 cycles of 15 s of denaturation at 95 °C, 40 cycles of 1 min of primer hybridization and elongation at 60 °C as well as 30 s of final elongation at 60°. CFX Maestro 4.0 Software (BioRad, Hercules, CA, USA) was used for the visualization and analysis of the amplified products. 

### 2.4. Statistical Analyses

Anthropometric data are shown as mean values ± standard deviation. To designate differences among experimental groups, the Student’s *t*-test was applied.

Two programs were used to perform all analyses: R (version 2.0-1, R Foundation for Statistics Computing, https://cran.r-project.org (accessed on 10 June 2020)) and IBM SPSS (PS IMAGO, version 27, SPSS, Inc., Chicago, IL, USA). The SNPassoc package for R was performed for single-locus analysis with four genetic models (codominant, dominant, recessive and overdominant) for the groups based on BMI and fat%. The models were constructed with respect to the minor allele. The influence of single alleles for all mentioned data divisions were calculated with Pearson’s χ2 test using STAT.package for R.

## 3. Results

According to the Student’s *t*-test, no statistically significant differences were found between the CON and OVER groups for the age and height traits (*p*-value was between 0.67 to 0.73 for BMI division and 0.84 to 0.86 for fat% division), confirming the homogeneity of the population. The groups differed statistically in terms of BMI, fat% and the following related parameters: weight, metabolic age, visceral tissue index, FFM, fat and water, with *p*-values ranging from 0.00 to 0.01 (Table 1).

The measured genotype frequencies did not significantly differ from the Hardy–Weinberg equilibrium expectations for *IL-6* (rs1800795), with p-values ranging from 0.14 to 1, and was not applicable for *IL-6* (rs1800796) and *IL6* (rs13306435) because of very low represented genetic variation and very low minor allele frequencies (Table 2).

Results of the association analysis between SNPs within the *IL-6* gene (rs1800795) and BMI values are given in Table 3, and for fat% in Table 4. No statistically significant result was found. The frequency of minor alleles in groups for rs1800796 and rs13306435 was too low to build a significant association model with any data divisions that were taken into account. In rs13306435 there was only one allele A, which was in the control group in all divisions. In rs1800796 only genotypes G/G and C/G were present, appearing with a big difference in frequency (87.2% to 12.8%), with the absence of the C/C genotype. For the BMI division, the frequency of the C/G genotype was very low (4.76%) in the OVER_BMI_ group and 16.87% in the CON_BMI_ group; for the fat% division it was 5.26% in the OVER_Fat_ group and 14.15% in the CON_Fat_ group (Table 5 and Table 6). 

## 4. Discussion

For many years obesity-related diseases, including a general pro-inflammatory state, insulin resistance and cardiovascular as well as multi-organ metabolic disorders have been known. The pro-inflammatory condition is defined as low-grade, chronic inflammation, resulting from multiple immune and hormonal impairments in adipose tissue (AT), including cytokine release [15]. IL-6 is a pleiotropic cytokine, considered as a “metabolic hormone”, involved not only in the immune responses but also in glucose, protein and lipid metabolism [3,16]. IL-6 concentrations in low-grade inflammation are elevated in plasma and adipose tissue; rs1800795 in the promoter of the *IL-6* gene, which regulates its transcriptional activity, has been the most extensively studied candidate for the association with pro-inflammatory responses and other obesity-related disorders. However, it has been reported that blood IL-6 concentration does not depend on this polymorphism alone [17].

Both positive and negative associations between obesity and *IL-6* polymorphisms in multiple loci have been reported [3,18,19,20,21,22]. Our study, reporting the lack of a relationship between obesity parameters (BMI and fat%) and *IL-6* polymorphisms rs1800795, rs1800796 and rs13306435 in a homogenous population poses an additional voice in this discussion.

According to a recent meta-analysis, rs2069845 and rs1800796 seemed unrelated to obesity, and minor alleles of rs1800795 and rs1800797 may have increased and decreased the risk of obesity, respectively [3]. Surprisingly, in this meta-analysis only 19 studies have been found eligible, and 14 of them dealt with rs1800795. Although the general conclusion from all 14 studies was that minor allele C of rs1800795 significantly increases the risk of obesity, the analyzed studies presented various results. This may be related to the fact that the analyzed populations involved adolescents and children of various ethnic groups, and that various obesity parameters and related traits were taken into consideration. Two issues arising from the studies considered in the above-mentioned meta-analysis seem particularly interesting and correspond to our results. Firstly, two studies were provided in Poland and neither of them revealed a clear relationship between rs1800795 and obesity [23,24]. Secondly, it has been mentioned that dietary factors may modulate the relationship between polymorphisms, including rs1800795 and obesity-related traits [25].

Pyrzak et al. examined a population of children, and the results were somehow controversial. In obese children, homozygous CC and C allele carriers were more frequent than in the control group, but on the other hand, these unfavorable phenotypes were protective against fat accumulation and lipid abnormalities. Moreover, metabolic syndrome signs were less frequent in this group. Thus, the authors concluded that polymorphism 174G>C (rs1800795) does not seem to be associated with obesity and with the incidence of metabolic syndrome in children. It must be mentioned that obese children were the patients of an endocrinology clinic, so several disorders affecting this population should be taken into consideration and might have influenced the results [23].

In the Adler et al. study, the population consisted of male and female volunteers over 55 years old, recruited during a visit to a general practitioner. The aim of this study was the investigation of the relationships between depression and obesity parameters as well as genetic traits, including *IL-1* and *IL-6* genetic polymorphisms. They found a correlation between depression and BMI as well as fat%, but did not find an influence of the studied polymorphisms. Due to the recruitment method, the population was largely heterogenous, and the life habits and other environmental factors might have been much more related to both depression and obesity than genetic polymorphisms were [24]. 

One study regarding rs1800795 included in the Gholani et al. meta-analysis has already mentioned the significant role of dietary factors [25], and there are more similar findings available [26,27,28,29]. Joffe et al. for the first time indicated that dietary fat intake modulates the relationship between the *IL-6* − 174 G>C (rs1800795) polymorphism and obesity as well as serum lipids in white South African women. The increased intake of n-3 polyunsaturated fatty acids (PUFA) decreased adiposity, and with an increasing n-6 to n-3 PUFA ratio adiposity increased in minor C-allele carriers [25]. This is in line with the previous findings regarding gene–PUFA interactions [26]. They have analyzed the PUFA content in erythrocyte membranes and revealed that the risk of obesity in C-allele (rs1800795) carriers decreases with increased fatty acid content, suggesting the possible beneficial effect of PUFA supply. The other study indicated that the C-allele of *IL-*6 - 174 G>C (rs1800795) was associated with increased postprandial fat oxidation. Furthermore, C-allele was protective against weight regain after weight loss [29,30]. Other diet-related factors involve the intake of polyphenol-rich cloudy apple juice, which reduced body fat depending on an rs1800795 polymorphism with a CC variant predisposed to lowering fat [28]. It has also been postulated that rs1800795, in combination with a flavonol-rich diet, has been associated with a lower risk of adenoma recurrence [27]. On the other hand, studies reported no dietary effects [31].

Taking the above into account, our study gave the unique opportunity to investigate the role of rs1800795 in a homogenous population, allowing us to avoid environmental factors that might have largely influenced the results of studies based on heterogenous populations. The group in our study consisted of young men, clinically and genetically healthy, without inflammatory disorders and with daily physical activity due to their military schedules. The group was also uniform regarding the environmental conditions. They were billeted in dormitories of the military college in Warsaw. All of them have undergone fitness tests during recruitment to the military college to ensure an appropriate level of physical fitness, and keeping this fitness was an essential part of their daily routine. They usually ate their meals in the university canteen, so their habitual diet did not vary and meals were balanced in terms of nutritional and energetic value, according to the recommendation for military professionals. 

Due to the lack of a relationship between *IL-6* polymorphisms (particularly the main candidate, rs1800795) and BMI as well as fat% in this homogenous group, it may be postulated that even if a genetic predisposition involves the *IL-6* gene, this effect is minor and can be avoided or at least markedly reduced by changes in lifestyle. 

Neither the polymorphisms examined in the context of obesity (rs1800795 and rs1800796) nor rs13306435, examined in the context of colorectal cancer [32], in which obesity poses a risk factor, were significant. Thus, it can be postulated that *IL-6* polymorphisms may be significant in severely affected patients, but not in low-grade obesity.

## 5. Limitations

The main limitations of this study were the small group and lack of severely obese individuals. However, these result from the recruitment of a homogenous group. The study design included the assumption of testing a homogenous group to avoid the influence of unpredictable environmental factors. That is why we have recruited only students that lived and ate at the Military University and had the same physical activity. We could only test volunteers, who were recruited via advertising and agreed to participate in the experiment. To maintain the homogeneity criteria, we could not add other subjects, and that is why the sample size was small and there were no severely obese soldiers.

## 6. Conclusions

It can be presumed that *IL-6* polymorphisms are not significantly involved in the development of low-grade obesity. Due to the lack of a relationship between *IL-6* polymorphisms and BMI as well as fat% in the studied group, it may be presumed that even if a genetic predisposition involves *IL-6* genes, this effect in individuals with low-grade obesity is minor, or can be avoided or at least markedly reduced by changes in lifestyle. 

## Figures and Tables

**Table 1 genes-12-01498-t001:** Anthropometry and body composition of the participants.

Group	All (*n* = 125)	CON_BMI_ (*n* = 83)	OVER_BMI_ (*n* = 42)	CON_Fat%_ (*n* = 106)	OVER_Fat%_ (*n* = 19)
Age	21.90 ± 1.65	21.87 ± 1.57	21.98 ± 1.79	21.92 ± 1.61	21.84 ± 1.87
Height (cm)	180.04 ± 6.36	179.87 ± 6.31	180.38 ± 6.43	179.99 ± 6.12	180.32 ± 7.53
Weight (kg)	79.35 ± 9.03	75.27 * ± 6.51	87.40 * ± 7.81	77.49 * ± 7.57	89.70 * ± 9.50
Metabolic age	19.22 ± 7.40	15.93 * ± 4.62	25.74 * ± 7.54	16.87 * ± 5.04	32.37 * ± 3.84
Fat (kg)	13.13 ± 4.36	11.06 * ± 2.40	17.22 * ± 4.47	11.82 * ± 2.85	20.46 * ± 4.05
Water (%)	60.66 ± 3.19	61.81 * ± 2.42	58.38 * ± 3.30	61.55 * ± 2.42	55.66 * ± 2.19
BMI (pts)	24.45 ± 2.21	23.24 * ± 1.24	26.84 * ± 1.69	23.89 * ± 1.70	27.56 * ± 2.12
Fat%	16.27 ± 3.85	14.63 * ± 2.68	19.51 * ± 3.76	15.13 * ± 2.79	22.64 * ± 2.49
FMI (pts)	4.04 ± 1.31	3.42 * ± 0.72	5.29 * ± 1.32	3.64 * ± 0.85	6.28 *± 1.15
Visceral tissue index	2.84 ± 1.71	2.02 * ± 0.90	4.45 * ± 1.79	2.32 * ± 1.09	5.74 * ± 1.65
FFM (kg)	66.22 ± 6.01	64.21 * ± 5.43	70.18 * ± 5.07	65.67 * ± 5.82	69.24 * ± 6.18

* Statistically highly significant difference. BMI—body mass index, FMI—fat mass index, FFM—fat-free mass, CON—control group, OVER—overweight group.

**Table 2 genes-12-01498-t002:** Frequencies for *IL-6* genotypes in ALL, OVER_BMI_, CON_BMI_, OVER_Fat_ and CON_Fat_ groups; probabilities that the genotype frequencies do not differ from Hardy–Weinberg expectations (HWE) and minor allele frequency (MAF).

SNP	MAF (%)	ALL	OVER_BMI_	CON_BMI_	OVER_Fat_	CON_Fat_
*IL-6* (rs1800795)	Allele C (45.60)	0.21	0.76	0.26	0.38	0.33
*IL-6* (rs1800796)	Allele C (6.40)	NA	NA	NA	NA	NA
*IL-6* (rs13306435)	Allele A (0.40)	NA	NA	NA	NA	NA

MAF—minor allele frequency, NA—not applicable.

**Table 3 genes-12-01498-t003:** Association analysis of the *IL-6* rs1800795 polymorphism with BMI.

		OVER_BMI_	%	CON_BMI_	%	OR	CI 95%		*p*-Value	AIC
Codominant	G/G	10	23.8	23	27.7	1.00			0.71	164.9
	C/G	23	54.8	47	56.6	1.13	0.46	2.75		
	C/C	9	21.4	13	15.7	1.59	0.52	4.92		
Dominant	G/G	10	23.8	23	27.7	1.00			0.64	163.4
	C/G-C/C	32	76.2	60	72.3	1.23	0.52	2.89		
Recessive	G/G-C/G	33	78.6	70	84.3	1.00			0.43	163.0
	C/C	9	21.4	13	15.7	1.47	0.57	3.78		
Overdominant	G/G-C/C	19	45.2	36	43.4	1.00			0.84	163.5
	C/G	23	54.8	47	56.6	0.93	0.44	1.96		
Alleles	G	43	51.19	93	56.02	0.82	0.47	1.44	0.55	
	C	41	48.81	73	43.98					

OR—odds ratio, 95% CI—confidence intervals, AIC—the Akaike information criterion.

**Table 4 genes-12-01498-t004:** Association analysis of the *IL-6* rs1800795 polymorphism with fat%.

		OVER_Fat_	%	CON_Fat_	%	OR	CI 95%		*p*-Value	AIC
Codominant	G/G	3	15.8	30	28.3	1.00			0.49	111.1
	C/G	12	63.2	58	54.7	2.07	0.54	7.90		
	C/C	4	21.1	18	17.0	2.22	0.45	11.08		
Dominant	G/G	3	15.8	30	28.3	1.00			0.23	109.1
	C/G-C/C	16	84.2	76	71.7	2.11	0.57	7.75		
Recessive	G/G-C/G	15	78.9	88	83.0	1.00			0.67	110.4
	C/C	4	21.1	18	17.0	1.30	0.39	4.39		
Overdominant	G/G-C/C	7	36.8	48	45.3	1.00			0.49	110.1
	C/G	12	63.2	58	54.7	1.42	0.52	3.89		
Alleles	C	20	52.63	94	44.34	1.39	0.66	2.97	0.44	
	G	18	47.37	118	55.66					

OR—odds ratio, 95% CI—confidence intervals, AIC—the Akaike information criterion.

**Table 5 genes-12-01498-t005:** Division of genotypes and alleles in the *IL-6* gene (rs1800796).

*IL-6* rs1800796	OVER_BMI_	%	CON_BMI_	%	OVER_Fat_	%	CON_Fat_	%	ALL	%
G/G	40.00	95.24	69.00	83.13	18.00	94.74	91.00	85.85	109.00	87.20
C/G	2.00	4.76	14.00	16.87	1.00	5.26	15.00	14.15	16.00	12.80
C/C	0.00	0.00	0.00	0.00	0.00	0.00	0.00	0.00	0.00	0.00
G	82.00	97.62	152.00	91.57	37.00	97.37	197.00	92.92	234.00	93.60
C	2.00	2.38	14.00	8.43	1.00	2.63	15.00	7.08	16.00	6.40

**Table 6 genes-12-01498-t006:** Division of genotypes and alleles in the *IL-6* gene (rs13306435).

*IL-6* rs13306435	OVER_BMI_	%	CON_BMI_	%	OVER_Fat_	%	CON_Fat_	%	ALL	%
T/T	42.00	100.00	82.00	98.80	19.00	100.00	105.00	99.06	124.00	99.20
A/T	0.00	0.00	1.00	1.20	0.00	0.00	1.00	0.94	1.00	0.80
A/A	0.00	0.00	0.00	0.00	0.00	0.00	0.00	0.00	0.00	0.00
T	84.00	100.00	165.00	99.40	38.00	100.00	211.00	99.53	249.00	99.60
A	0.00	0.00	1.00	0.60	0.00	0.00	1.00	0.47	1.00	0.40

## Data Availability

The data presented in this study are available on request from the corresponding author. The data are not publicly available due to ethical reasons.

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
