# Peer review of "IL-6 Polymorphisms Are Not Related to Obesity Parameters in Physically Active Young Men"

_genes, 2021, doi:10.3390/genes12101498_

Round 1

Reviewer 1 Report

The manuscript by Dr. Paweł Cięszczyk and colleagues provides evidence that IL-6 polymorphisms may be significant in severely affected patients, but not in the obesity of low grade. This is an interesting study about a “metabolic hormone” involved in obesity which is a global health problem.

It is expected that other factors are founded by the results of further genetic studies.

A few comments for considerations:

  1. Table 1: This table is separated and not clear.

  1. All results should be written in the result section. The description is not enough.

  3. What is the concentration of IL-6 in plasma of these subjects?

Author Response

We would like to kindly thank the Reviewers for their opinions, suggestions and comments for our  article. All comments have been addressed, and changes in the text have been made. Please, find the point by point answers below:

  1. Table 1: This table is separated and not clear.

Sorry for the mistake – the table was not indicated in the text. It has been corrected. The table presents the obesity related traits that formed the basis for the differentiation of the groups and the traits (age and height) that did not differ, confirming the homogeneity of population (also added). The table was also moved entirely to the next page so that the rows would not be separated.

  1. All results should be written in the result section. The description is not enough.

We added additional Tables (5 and 6) with data to the fragment where there was only a description.

  1. What is the concentration of IL-6 in plasma of these subjects?

We would love to have these results, unfortunately, only the non-invasive procedures were allowed, so only buccal swabs could be taken and we couldn’t measure any biochemical parameters in plasma. Only DNA concentration was measured prior to genotyping (~20ng/ul).

Reviewer 2 Report

Authors of the present paper aimed at investigating the most commonly studied polymorphisms of IL-6 in a group of healthy young males, homogenous in terms of environment and lifestyle (military professionals), differentiated by body mass index – BMI and the percentage of fat.

The article presents quite interesting results, but despite this, there are some issue that authors should address.

The main limitation of this study is the sample size that is very small for a genetic study. The associations investigated in previous studies used larger sample sizes.

  • The study population is poorly defined.
  • No estimations of ideal sample size was performed
  • Two people of similar BMI could have widely different body composition and %fat (as you noted); BMI<25 may still have high %fat. So waist circumference is better than BMI
  • The authors should be cautious about making causative statements. All of the factors mentioned reasonably play a role in obesity. So you can’t write “It can be presumed that IL-6 polymorphisms may be significant in severely affected patients, but not in patients with obesity of a low grade. Due to the lack of relation between IL-6 polymorphisms and BMI and fat% in the studied group, it may be presumed that even if a genetic predisposition involves IL-6 genes, this effect is minor and can be avoided or at least markedly reduced by the lifestyle” (end of Abstract, Conclusion). You don’t have severely obese subjects.

Author Response

We would like to kindly thank the Reviewers for their opinions, suggestions and comments for our  article. All comments have been addressed, and changes in the text have been made. Please, find the point by point answers below:

  • Authors of the present paper aimed at investigating the most commonly studied polymorphisms of IL-6 in a group of healthy young males, homogenous in terms of environment and lifestyle (military professionals), differentiated by body mass index – BMI and the percentage of fat.

        The article presents quite interesting results, but despite this, there are

         some issue that authors should address.

  • The main limitation of this study is the sample size that is very small for a genetic study. The associations investigated in previous studies used larger sample sizes.

We do agree that the population is very small for genetic study. However, our aim was to test the homogenous population to eliminate as many environmental factors as possible. Therefore we have selected military professionals who live together, eat the same food and have very similar daily schedule. Unfortunatelly, we could not test all military professionals (at certain age), but only volunteers. It was impossible to collect more samples while still having a homogenous group.

  • The study population is poorly defined.

More details have been added in the section 2.1 Participants

  • No estimations of ideal sample size was performer

 MAF for IL6 (rs1800795) was assumed at the value 0.44 for Europe ( 0.46 represented among our respondents), according to Statistics Poland the prevalence of overweight and obesity among men in respective age was at the level of 0.45

The study design included the assumption of testing homogenous group to avoid influences of unpredictable environmental factors. That is why we have recruited only students that lived and ate at the Military University and had similar physical activity. We could test only volunteers, who were recruited via advertising and agreed to participate in the experiment. To maintain the homogeneity criteria, we could not add other subjects and that is why the sample size was small.

  • Two people of similar BMI could have widely different body composition and %fat (as you noted); BMI<25 may still have high %fat. So waist circumference is better than BMI

We do agree, therefore we have measured more parameters (given in table 1), including FMI and FFM. In our study there were no paticipants with low BMI and high fat%, but there were soldiers with high BMI and low fat%. Unfortunately, we did not measure waist circumference, so we cannot add such data. However, we believe that multiple antropometric data justify the selection of groups.

  • The authors should be cautious about making causative statements. All of the factors mentioned reasonably play a role in obesity. So you can’t write “It can be presumed that IL-6 polymorphisms may be significant in severely affected patients, but not in patients with obesity of a low grade. Due to the lack of relation between IL-6 polymorphisms and BMI and fat% in the studied group, it may be presumed that even if a genetic predisposition involves IL-6 genes, this effect is minor and can be avoided or at least markedly reduced by the lifestyle” (end of Abstract, Conclusion). You don’t have severely obese subjects.

Thank you for this comment, indeed we did not have severely obese subjects and they were not even expected in young military professionals population. The sentences in abstract and conclusions have been corrected accordingly.

Round 2

Reviewer 1 Report

Authors have fully responded the reviewer's comments and requests. The revised manuscript is a strong candidate for publication in Genes.

Reviewer 2 Report

The paper was greatly improved, but still the main problem is the sample size that is very small for a genetic study, for a SNPs study.

Please reconsider after enrolling more subjects